# Exploration of mentor and mentee perspectives of a mentored clinical practice programme to improve patient outcomes in musculoskeletal physiotherapy

**Aled Williams** [1]* *, **Ceri J. Phillips** [2]*, **Alison Rushton** [3]*

**1** School of Healthcare Sciences, College of Biomedical and Life Sciences, Cardiff University, Cardiff, Wales, United Kingdom, **2** Swansea Centre for Health Economics, Swansea University, Swansea, Wales, United Kingdom, **3** School of Physical Therapy, Western University, Elborn College, London, Ontario, Canada

☯ These authors contributed equally to this work.
* WilliamsA146@cardiff.ac.uk

**Data Availability Statement:** All relevant data are included within the paper.

**Funding:** The authors received no specific funding for this work.

## Abstract

### Background

A recent randomised controlled trial has demonstrated the impact on practice of an educational programme for clinicians. Mentored clinical practice in musculoskeletal physiotherapy resulted in clinically significant improvements in both physiotherapist performance and patient outcomes. The objectives of this study were to explore mentor and mentee perceptions of a mentored clinical practice programme, in order to identify key factors in the process to improve patient outcomes.

### Methods

Employing a case study design of a mentoring programme that led to improved patient outcomes, mentored clinical practice was explored from multiple perspectives using a grounded theory strategy of enquiry to derive a theory of mentored clinical practice grounded in the views of the participants. Semi-structured interviews with a purposive sample of mentors and mentees were employed along with qualitative observations of mentored clinical practice. Data analysis and collection were concurrent, with analysis an iterative process deriving inductive analytical categories from the data through constant comparison.

### Findings

Highly informative themes of how the complex interaction between mentor, mentee, patient and environment worked successfully were identified from the data. The mentors' knowledge, additional perspectives, critical analysis and facilitatory style were enabling factors, as were mentees' motivation, openness to criticism and commitment to reflect on practice. Themes around potential threats to the mentees' development were also identified. Overloading or contradictory feedback and lack of relationship with mentees were barriers that mentors could bring; fear, defensiveness, routine working, people-pleasing and lack of

**Competing interests:** The authors have declared that no competing interests exist.

experience were potential mentee barriers. A model emerges from the data demonstrating how these themes interact, providing guidance to mentors and mentees to optimise the effectiveness of mentored clinical practice.

## Conclusion

This study provides a sound basis for future mentored clinical practice, producing a model from key themes from a case study where impact on clinician performance and patient outcomes are established.

## Introduction

Demand over several decades has called for the evaluation of clinician education using the 'gold standard' of patient outcomes [1–8], but this is acknowledged as difficult to achieve [9–15]. A recent low risk of bias randomised controlled trial (RCT) [16,17] demonstrated the impact of an educational intervention of 150 hours of mentored clinical practice (MCP), with musculoskeletal (MSK) patients being treated by physiotherapists who had received MCP more than four times more likely to achieve a clinically significant outcome than those treated by physiotherapists who had received usual training. In answering the calls to evaluate clinician education with patient outcomes, clinical mentorship to develop clinical reasoning skills to a higher level was shown to improve clinical effectiveness. This naturally raises questions about the content of such an effective educational intervention in order for it to be applied to clinical practice for the benefit of future clinicians and their patients.

The MCP intervention was informed by international MSK educational standards [18] set by the International Federation of Orthopaedic Manipulative Physical Therapists (IFOMPT), a subsidiary of World Physiotherapy, which in turn is a member of the World Health Organisation. These standards are set for the governance of MSK physiotherapy education and inform MSK curricula at Masters level [19] and map onto advanced clinical practice frameworks [20]. IFOMPT accredited programmes utilise clinical placements for mentee clinicians to be guided by mentors in order to develop and apply high levels of clinical reasoning, advanced use of knowledge and personal characteristics to their clinical practice [21]. Graduating from IFOMPT accredited programmes correlates with superior patient outcomes for graduates [22, 23], who testify to the MCP component being most influential on practice change [24, 25]. Such are the perceived benefits of this MCP approach, a recent study protocol is exploring the efficacy of novel ways to apply the benefits of MCP to practice [26]. Knowing that the mentorship led to clinical effectiveness raises questions around the perceptions of the intervention to understand its characteristics contributing to its success.

### Objectives

The objectives of this study were to explore mentor and mentee perceptions of a work based MCP programme, to observe the delivery of MCP in real time and in context and to identify key factors in the process of MCP to impact patient outcome for applications for future practice and research.

## Methods

### Study design & methodology

A multiple methods study design of semi-structured interviews with mentors and mentees and qualitative observations of the mentoring process was employed using multiple stages of data

collection across multiple populations (mentors and mentees) [27]. This design was developed on the basis of specific strategies and the standpoint of the researchers.

Employing a case study design of a work-based MCP programme that led to improved patient outcomes [17], qualitative methodology was chosen with the primary aim of exploring MCP from multiple perspectives. The underpinning epistemological stance was pragmatic, being concerned with applications (what works in obtaining improved patient outcomes) and solutions to problems (the lack of research linking education to patient outcomes) [28]. This pragmatic stance was congruent with a grounded theory strategy of enquiry, for the researcher to derive a theory of MCP and the actions and interactions of the mentor, mentee and patient grounded in the views of the participants [28].

The nature of the case study [17] was determined by being embedded within stepped-wedge design methodology for the RCT, meaning that the MCP intervention was rolled out across different clusters (physiotherapy departments) arranged geographically into North, South and West. This allowed staggered sections of both methods of data collection and assisted the iterative data analysis process (see Fig 1).

**Participant selection.** Purposive (maximum variation) sampling from the group of 16 mentee physiotherapists and the 9 mentors from the case study [17] provided the interviewees for this study. All were qualified physiotherapists practicing in 6 musculoskeletal outpatient physiotherapy departments in a single NHS organisation and were approached to participate via email and subsequent face-to-face discussion. This sampling approach was taken in order to obtain a wide range of diverse perspectives on MCP in order to identify shared patterns across cases [29]. There was one dropout from the study (non-work related stress) during the qualitative data collection and no refusals to participate. Saturation of themes occurred with a sample of 6 observations and interviews with 12 mentees and 3 mentors. Demographic data from the sample showed that 11 of the 12 mentees were employed at band 6 level and one at band 7 on the NHS Agenda for Change pay system [30] (i.e. all were in senior physiotherapist roles) at the start of the study. The gender balance of the sample was 5 male to 7 female; all 3 mentors interviewed were female. The mean years' experience in the MSK field for mentees was 7.5 years (range 3–21 years). All mentors were members of the UK member organisation

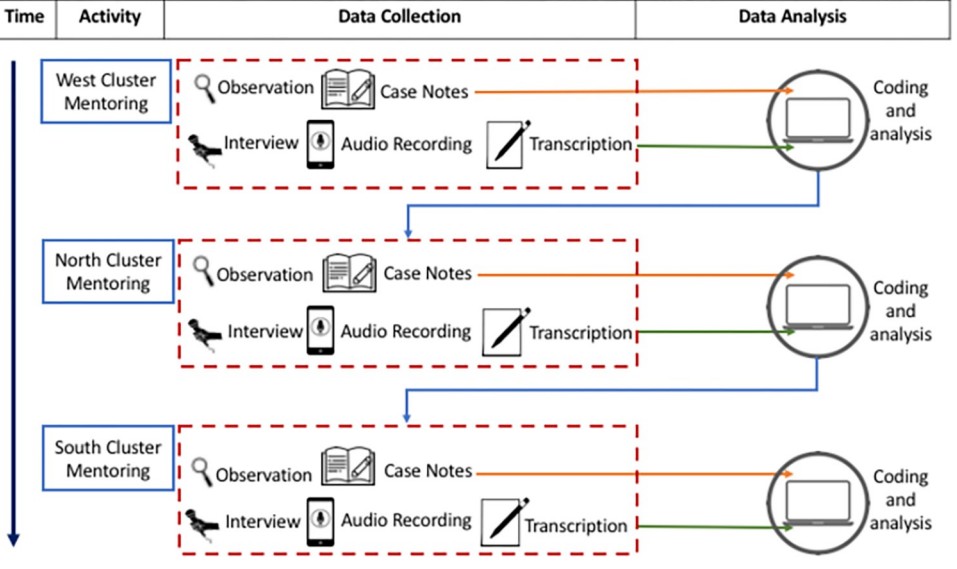

**Fig 1. Study design.**

of IFOMPT, the Musculoskeletal Association of Chartered Physiotherapists (MACP). Having undertaken postgraduate study and reached a recognised standard of excellence in musculoskeletal physiotherapy, the mentors also had experience in delivering mentored clinical practice for IFOMPT accredited postgraduate programmes.

**Data collection & settings.** Data collection took the form of participant observations of the MCP process and semi-structured interviews with mentors and mentees. These took place in the usual work contexts of the participants, namely the Outpatient departments of the MSK Physiotherapy Service of a large NHS organisation.

*Participant observations during MCP*. Qualitative observations of the MCP process allowed the lead researcher (the lead author of this paper) to position himself in the process of the MCP and take structured field-notes on the behaviour and activities of participants and their mentors [31, 32]. The lead researcher was able to record the dynamic between mentor and participant, as well as directly witness the intervention in progress in the participant's usual place of work. Qualitative observational research relies on the researcher as the instrument documenting the world being observed [33]. A simple observational protocol was used for recording data from observations of MCP sessions. Descriptive notes included reconstructions of dialogue, descriptions of the physical setting and accounts of activities while reflective notes included the lead researcher's thoughts, feelings and ideas as researcher [31]. The initial notes and observations were used as a prompt to write up full field notes as soon as possible after the observations [32]; NVivo10 software was subsequently utilised to assist in the data management.

*Semi-structured interviews*. Interviews were conducted by the lead researcher at the participant's usual place of work at their earliest convenience following MCP, ensuring an applicable context [34]. Only the lead researcher and participant were present during interviews which were digitally recorded, transcribed and imported into NVivo10 software for concurrent analysis. Interview lengths varied but typically lasted around 1 hour; no repeat interviews or participant transcript checks were used. The interview prompts were developed and expanded from a professional development interview used by researchers investigating expertise in musculoskeletal physiotherapists [35].

*Settings*. The MCP took place in NHS outpatient physiotherapy departments with the mentor observing the mentee delivering usual care to the patient: taking the patient's history, performing a physical examination, as well as delivering treatment by way of communicative and instrumental strategies (e.g. prescribing exercise, applying manual therapy techniques). The ethical issue of additional clinicians being present who were not directly involved with the patient's care was addressed and informed consent was gained from all patients. At the end of the patient history the mentee would move to an adjacent room and summarise his/her reasoning to the mentor, interpreting key findings to justify a focus and plan for the physical examination with the mentor asking questions and giving guidance. In every observation, the time after the patient had left was where the greatest volume of discussion took place. The mentee was given some structure by the mentor to frame their thinking and was expected to reflect on the patient's experience, perspective, diagnosis and management and on their own reasoning, decision-making and performance.

**Research team and reflexivity.** The lead researcher had credibility and credentials for this process, being an experienced mentor for masters level students and having previously undertaken a masters degree which involved a clinical placement module of mentored clinical practice. At the time of the interviews, the lead researcher was working within the same NHS organisation with an established relationship with the participants as a work colleague. Participants were aware of the lead author's motives and reasons for the study; the study was part of the lead author's PhD research. Participants changing their answers or behaviour to give the

lead researcher what they might think be desired responses was considered [36] and a trusting relationship with emphasis on confidentiality and anonymity was established.

**Ethical considerations.**   Informed written consent was gained from all eligible mentors and mentees in the sample. Mentors and mentees were interviewed individually rather than in focus groups in order to maintain anonymity. Care was also taken that any quotes would not be attributable to the physiotherapists involved, through use of participant pseudonyms and care being taken with data displays not to include details that could lead to identification of the participants. Ethical approval for the study was sought and obtained from the South East Wales Research Ethics Committee C (ref: 12/WA/0078).

**Data analysis.**   Data analysis and collection (participant observations and interviews) were concurrent, with interim analysis of data already gathered allowing the lead researcher to shape ongoing data collection through question refinement and pursuing emerging themes along with exploration of deviant cases [37]. Data analysis was an iterative process with inductive analytical categories derived from the data through constant comparison [38]. Data coding was an inclusive process [37] with cross indexing of data displaying multiple themes. NVivo10 software was utilised to assist a systematic and coherent approach with a second person cross-checking the lead researcher's codes; inter-coder agreement was above the 80% minimum set for good qualitative reliability [39]. The lead researcher took a critically reflective stance through the data collection and analysis process, with participants checking and verifying the analysis [40]. Saturation was determined as the point of cessation for each method of data collection [41, 42].

**Quality of data.**   The COREQ statement [43] highlights important elements to disclose for interview research. Each theme is illustrated with relevant participant quotations with major themes developed and discussed as well as minor themes and divergent cases [40]. The trustworthiness of the findings appears good: the integration of themes across qualitative observations and mentee and mentor interviews gives credibility to the findings [44]. The congruence of these themes with those of previous qualitative literature on physiotherapy expertise and masters level practice support the truthfulness of the findings [44]. Furthermore, the standpoint of the lead researcher, being outside academic institutions where this type of intervention is employed, as well as employing measures to reduce researcher bias supports the confirmability of the findings [28]. The presence of the interview protocols and prompts supports the dependability of the study's findings as consistent and repeatable [28].

## Findings

**Emerging themes.**   Multiple themes were identified from the integrated interview and observation data which were highly informative of what was happening in the MCP process. In addition to how the complex interaction between mentor, mentee and patient worked successfully to develop the mentees in their usual clinical contexts, potential threats to the mentees' development were also identified. These themes are summarised and represented in Table 1 below, with a notional ranking from the lead researcher in perceived order of importance (determined from the balance of time and emphasis applied by the participating mentors and mentees in the interviews).

The emerging themes highlighting what supported the mentee's development will be detailed first, followed by those that were potential threats or barriers.

**Supports to MCP.**   *Person 1*: *The mentor–perspective, challenge and facilitation from an experienced and supportive partner.* ***Theme 1: Mentors bring additional perspectives:*** In its essence, MCP provided an additional perspective on the therapist-patient interactions, with the mentor being able to observe and critique different aspects of the mentee's practice, thus providing an angle that the mentee would struggle to produce independently. Ostensibly, the

**Table 1. Themes emerging from qualitative data.**

| Main theme categories and sub-themes | Mentee interviews | Mentor interviews | Observations |
|---|:---:|:---:|:---:|
| **Mentor Contribution** | | | |
| • Additional perspective | √ | √ | √ |
| • Critical analysis and challenge | √ | √ | √ |
| • Constructive, facilitatory style | √ | √ | √ |
| • Clinical & mentoring experience | √ | √ | √ |
| • Support, empathy, partnership | √ | √ | |
| **Mentee Contribution** | | | |
| • Motivation | √ | √ | √ |
| • Openness | √ | √ | |
| • Reflection | √ | √ | |
| • Practice knowledge | √ | | |
| **Patient-Centeredness of process** | | | √ |
| **Environmental Contribution** | | | |
| • Comfort / familiarity | √ | | |
| • Immediate application | √ | | |
| • Peer support | √ | | |
| • Prior relationship with mentor | √ | | |
| **Mentee barriers** | | | |
| • Defensive attitude | √ | √ | |
| • Routine working | √ | √ | |
| • People pleasing | √ | √ | |
| • Lack of knowledge / experience | √ | √ | |
| • Fear of exposure | √ | √ | |
| **Mentor barriers** | | | |
| • Lack of prior relationship | √ | √ | |
| • Overloading / directive feedback | √ | | |
| • Differing emphases | √ | | |
| **Environmental barriers** | | | |
| • Reduction of reflective time | √ | | |
| • Peer presence | √ | | |

clinician is 'multi-tasking' cognitively, seeking to listen to, understand and engage the patient, while generating and testing hypotheses, exploring potential patterns that the patient may fit and at the same time develop appropriate management strategies. MCP supplied an independent and experienced observer watching the assessment real-time and giving immediate feedback:

> **Mentee 2**:...*because they're sitting and watching the whole process... they're able to analyse the situation sometimes almost better than you are because they're...they have got the benefit of not having to... have the interaction as well as doing the thinking at the same time.* [476–480]

Each mentee had multiple mentors, which amplified these additional perspectives, as the mentors brought different thinking, qualities and skills to the process:

> **Mentee 3:** *I think they all worked very differently–which was good–because . . . their thought processes–they're all quite different. So, it was interesting 'cause they have very different kind of skills. . .* [524–533]

> ***Theme 2**: Mentor brings critical analysis and challenge***: The challenge to their practice was a theme picked up by several of the mentees, with the mentor providing a greater challenge than what the mentee could bring through self-reflection:

**Mentee 4:**...*you may have come out thinking "I thought that went quite well" and then suddenly to find there's a load of things that they didn't feel perhaps went as well...* [528–531]

It is worth noting that the challenge was not on the subjective, superficial basis of the mentor critiquing on the basis of what they liked or how they would approach the patient. Rather, the challenge was on the more objective, deeper level of evidence, pushing the mentees for evidence to justify their reasoning and decision-making:

**OBSERVATIONAL NOTES (O3):** *Mentor focus: reasoning. Emphasis on justification for decisions. Strongly interpretive.*

**Mentee 2:**...*it's the challenging of your... evidence, of what evidence have you got for... your hypothesis that you've just told me? And it's... making you think about that and making you just be more specific in the testing that you would do to get your answers.* [473–476]

This challenge to practice was delivered in the context of an honest and open relationship between the mentor and mentee. Mentors were cognizant of the potential to affect the mentee in a negative manner, especially in light of MCP occurring in the usual workplace and that their relationship would continue beyond the process. Mentors would also exchange places, inviting the mentee to observe them and to critique and challenge their practice to enhance the openness of this relationship. Mentees, appreciated the honesty and challenge to their practice, albeit after occasionally becoming defensive in difficult discussions with mentors:

**Mentee 5:**...*we talked openly, we were very honest, both very honest I really appreciated that time, with (my mentor)... I think I really appreciated that rather than just brushing that under the carpet.* [337–342]

**Mentor 2:** *I'd get them to watch me and analyse me. I'm quite open about my mistakes because I think that then opens up opportunities for them to not feel quite so vulnerable or if you've shown a degree of vulnerability they are then quite happy to do that.* [86–89]

Feedback from mentors was both balanced (positive and negative) and strategically prioritised in response to the mentee's learning styles and needs, with a focus on what would elicit maximum meaningful change to the mentee's practice first. Specifically, this focus was achieved by the mentors balancing the critique with elements of reflection and direction:

**Mentor 3:** *I think in the first instances you would try to understand their major weaknesses in terms of their reasoning and their major reasoning errors and try and focus on those to begin with.* [19–23]

**OBSERVATIONAL NOTES (O4):** *Mentor doesn't just relive the whole assessment; mentor puts focus onto key points and allows mentee to interpret.*

***Theme 3: Mentor uses a constructive, facilitatory style:*** A clinician could perceive the critique of their practice knowledge and behaviour as a negative experience. However, the mentors questioning approach, facilitating the mentees' own self-reflection and analysis, was embraced positively by the mentees as they were able to see for themselves where they were weak and how they could improve:

**Mentor 2:** *I guess you want to. . . try and help develop them to really be driving the reflection and analysis rather than just banging out a list of 'you did this right and you did this wrong', so it's facilitating self-reflection. [18–21]*

This reflective approach did not mean that directive feedback was absent; specific instruction and guidance was seen across all observations, but typically occurred at the end of the reflection section of the mentor-mentee discussion, once the mentee's reflection had concluded:

**Mentee 7:** *when [the fellow mentee] and I had come up with as much things as we could think of he would say "Well actually have you thought about X, Y, Z? Have you got any concerns about. . .? Is there anything you need to be careful about. . .?" Things that we hadn't thought of, so that was good. [295–297]*

One of the most striking reflections was the lack of emphasis on hierarchy in the mentor-mentee relationship. Undoubtedly, such a hierarchy existed, with the mentors being senior clinicians possessing M level qualifications, but mentors and mentees suggested that playing on any hierarchy could be detrimental to the process:

**Mentee 2:** *it is the collaboration because it's sort of having somebody to. . .in a way. . . guide your thinking. It's not just about them saying 'this is how I do it and this is the best way and you must learn it this way' because that's not what it's about. . . it's the other person. . .making you talk about 'Why? Why do you think that?' [108–112]*

***Theme 4: Mentor brings experience from clinical practice and previous mentoring:*** While the knowledge and experience the mentors possess are qualifying characteristics for the role of mentor, two specific applications of the benefits of this knowledge and experience emerged from the data. Firstly, the patient receives not only the reasoning, knowledge and experience of the mentee, but also that of a knowledgeable and experienced mentor, specifically for their benefit:

**Mentee 2:** *one of the mentors. . .I was working with. . .he's very knowledgeable about a lot of different concepts and you know, different papers and. . .and so can. . . use that information to help reason through a particular problem with a patient. [92–95]*

Secondly, the mentor brings knowledge and experience of delivering MCP. Being comfortable and familiar with the process allows the mentor to recognise patterns of reasoning and behaviour in the mentee and to display flexibility in responding to the learning needs and styles of the mentee:

**Mentee 5:** *Both mentors [were] very willing to be quite fluid, both quite happy to work out what was going to work out best for me and for [fellow mentee]. We worked out that, if you feel like I can change something there and then, I'd probably rather know about it there and then. . . so that fluidity was a really nice experience. . . [328–335]*

***Factor 5: Mentor brings support, positivity and empathy through an ethos of partnership:*** Encompassing every aspect of what the mentor brought to MCP was a high level of support. The mentors were able to identify with the mentees having been through a similar process themselves as part of their M level learning and this empathy was an integral feature:

**Mentor 2:** *I think one of the biggest things that's most helpful in any kind of teaching is a reasonable identification with the individual as in 'I've been in your situation and I know what it's like'. . .I have been in that process and it is stressful and it's hard and you're trying to change things. [264–268]*

**OBSERVATIONAL NOTES (O3):** *The mentoring process itself is discussed explicitly, not just implicitly. Mentor shows empathy and delivers constructive feedback.*

The empathy of the mentors made them cognizant of the levels of fear and anxiety that mentees might be experiencing going into the process and they demonstrated skill at overcoming this:

**Mentee 9:** *Initially I was worried about thinking, you know, that really senior physios were coming to watch me work and I've always had an element of under confidence about my clinical practice anyway but they dispelled that immediately and they were completely non-judgemental and totally supportive of me through the process which is absolutely amazing. [981–986]*

Responsibility and care were taken by the mentors for confidentiality, mindful that undermining this would damage the relationship and lead to a reduction in the openness of the mentee to be honest in discussing their areas of weakness for development and ultimately reduce the impact for change of the whole process:

**Mentor 1:** *it should be a very personal relationship so. . . what you discussed and things it should be felt by the mentee that it doesn't go any further and I think that's an important part of the mentoring process and it has to be constant. . . [63–69]*

The desired effect of this supportive environment from the mentor's perspective was achieved in that it did allow the mentees to optimise their performance and also to encourage openness and honesty between the mentor and mentee.

*Summary.* Pulling these themes together, a picture of what effective mentors bring to the process of MCP to impact patient outcomes begins to emerge (Fig 2).

*Person 2: The mentee: A motivated, open reflector, building on previous knowledge and experience.* **Theme 1: Mentee brings motivation for practice change:** Fundamentally, for the mentee to engage with MCP, there was a requirement for the mentee to see the need–and have a desire–for the mentoring inputs described above. All participants were volunteers and so motivation was implicit; however, consideration is given to applying MCP outside of this study, in helping the mentee overcome their fear and working through the challenging nature of the process:

**Mentee 10:** *Even though I get really scared about things and very nervous about doing things and exposing myself, I still jump in with two feet and want to learn and I think that approach is helpful in these mentoring sessions. . . [281–283]*

The mentees were motivated by the unique level of support from an experienced mentor consistently delivering feedback which was bespoke to their practice. This motivation was centred around a desire for practice change, rather than continue to work in a routine way:

**Mentor 3 (speaking about one of her mentees)—***I think that when we started to point out areas that could be improved upon, he was quite keen to. . . make changes. . . he knew that he wanted to change and improve. [256–259]*

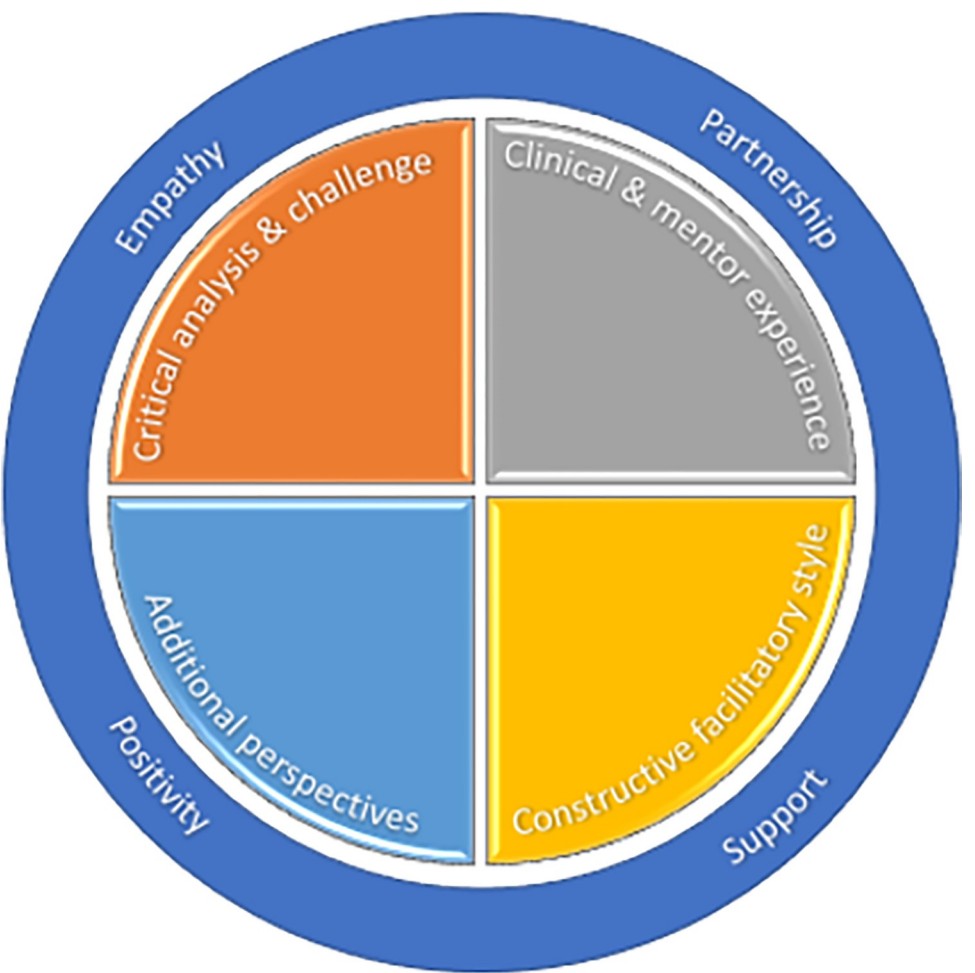

**Fig 2. Emerging model of MCP–mentor themes.**

*Theme 2*: *Mentee is open to receive constructive criticism*: The additional perspectives of the mentors brought real time feedback on the mentee's practice, with strengths and weaknesses identified and addressed. An openness to receive such critical challenge and analysis was identified by the mentees as a crucial factor for their engagement:

**Mentee 9:**. . .*they weren't criticisms they were suggestions and for me, I decided at the very beginning there was no point doing it unless I was going to be open like that because otherwise it was a pointless exercise. . . [958–961]*

This openness to the mentor's input and critique was seen to be effectively the hinge on which the process would succeed or fail in having a far-reaching impact on the mentee's practice. From a mentor perspective, a degree of mentee confidence is required—to invite such critique and a confidence in the process itself. The opposite is seen to stymie the impact of the process on practice:

**Mentor 1:** *Sometimes having a confidence issue can be the thing that really is frustrating and can perhaps hold people back that they are too afraid to tell you what they are thinking and*

*they don't open up to you and so ideally you want them to have the confidence in the process I suppose otherwise it stops it short really. [115–118]*

While a degree of inherent confidence was helpful in assisting such openness, a conscious choice by the mentee to expose their reasoning to the mentor was still required. This was assisted by reassurance from the mentors that critical appraisal was not a personal judgement but rather a necessary means to the end of practice change:

**Mentee 9:** *the very, very first thing that [the mentors] said to me independently was that they weren't there to judge me and from that point I made the decision that every other person has to make... 'OK I'm going to take that, I'm going to not be stressed about the fact that you're going to see things in me operating in ways which aren't going to be that good' so that's what I chose to do and that was the most significant thing for me. [932–937]*

***Theme 3: Mentee is committed to reflect on practice:*** What was observed consistently across MCP sessions was a significant amount of the time spent with the mentee reflecting on their practice, with the mentor emphasising the priority of this with their questioning:

**OBSERVATIONAL NOTES (O1):** *Mentee is doing most of the talking. Mentor performs role of facilitator of reasoning, asking questions to provoke/aid thought.*

*Mentor invites reflection with dual focus–the patient and the mentee. Mentor responds to direct question by directing back to mentee to reflect and reason.*

The basis for so much of the questioning being directed at generating mentee reflection was underlined by the mentors' expectations of this being a fundamental responsibility of the mentee, in order to apply learning to practice eliciting practice change. This cognitive "heavy" process requires time:

**Mentor 3:** *I think it's quite important that they take on the information that you have reflected upon together and they integrate it and practice it daily, so they can build on it for the next time, rather than saying the same thing time and time again because they are perhaps too busy to stop and think about it. [51–57]*

An interesting perspective from the mentors was that reflection was more highly valued than performance and that the mentors' facilitatory style is about developing skills not just for that immediate patient, but for future patients:

**Mentor 2 (speaking about what is important during MCP)–**...*ability to self-reflect. If you've done a shocker of an assessment, but at the end of it you can say "I know that was shocking because of X, Y, Z; if I did it again, I would do this, this and this". If they can articulate that process, then... that would be excellent... You want someone who's assured and comfortable in themselves and have a fair analysis of self-reflection so reflecting the good as well as the bad. [65–72]*

***Theme 4: Mentee brings previous knowledge and experience:*** In addition to the mentors' own knowledge and experience, the mentees each saw their own unique contributions to the process of their personal attributes, skills, experience, knowledge and learning as foundations on which to build:

**Mentee 6:** *The previous mentoring, the volume of patients seen over the years, that pattern recognition and that scheme of knowledge and patient experience that I've had in the past give me a good foundation to then go and take this a step further with the mentoring. [249–254]*

Belonging to a department where there was already a culture of observed practice meant that mentees felt less threatened and surprised by the types of questions being asked and more relaxed with the process as a whole:

**Mentee 11:.** *. .there is an element of being used to that process to an extent anyway so as I said I think I'm pretty relaxed being mentored. . . had I been brought into this process having not experienced it at all, to be totally honest I'm not sure if I'd have coped with it at all. [200–205]*

Bringing these themes together a picture emerges of what mentees bring to MCP for it to be effective for practice change and so to impact patient outcomes (Fig 3).

However, this model still lacks any representation of the third person involved in the process who is integral to its success.

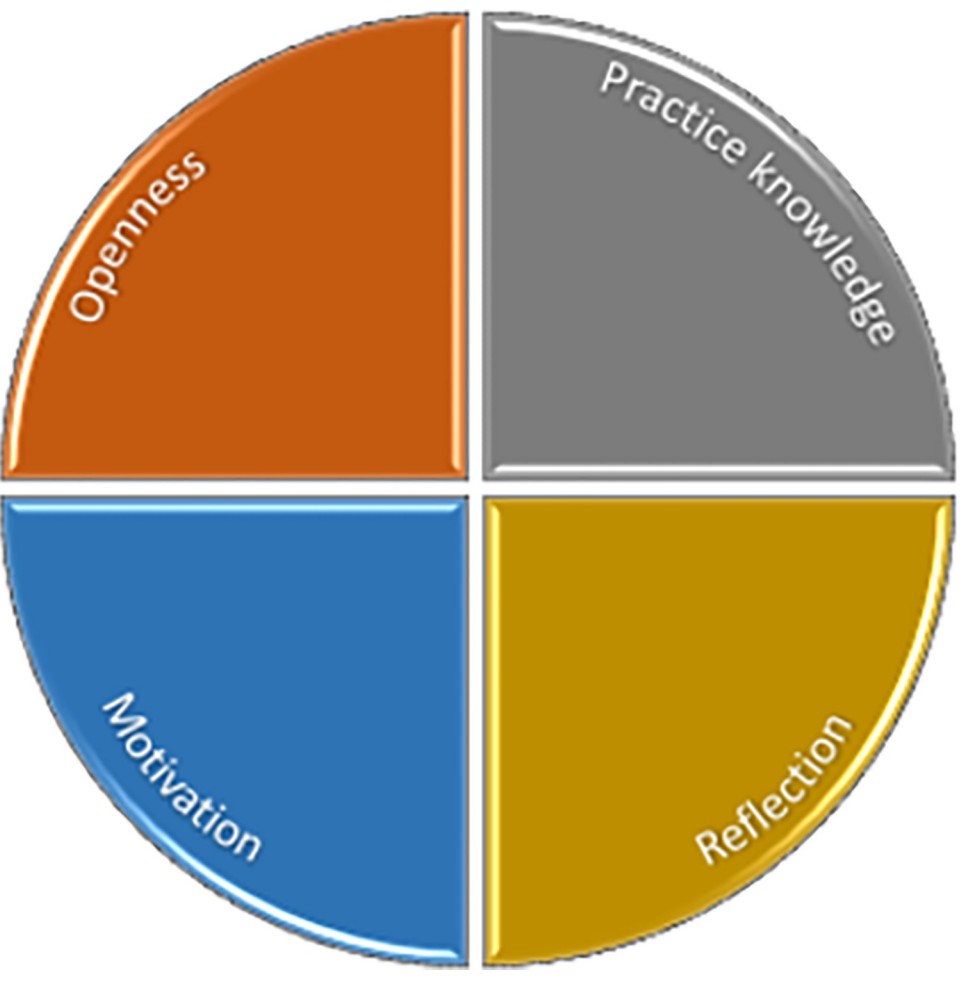

**Fig 3. Emerging model of mentoring–mentee themes.**

*Person 3*: *The patient*: *A respected beneficiary of holistic and reasoned care, performing an essential role in the process*. Throughout every observation each patient was respected and prioritised. Rather than being a "vehicle" by which the mentee could develop, the patient was at the centre of the interaction. This was seen in the care taken in consenting the patient, to individual aspects of the interaction–for example, asking permission to discuss reasoning with the mentor. At each point the patient was kept informed of the process, given the opportunity to withdraw consent and any discussion that did take place was discussed and interpreted in layman's terms for the patient.

**OBSERVATIONAL NOTES (O5):** *Patient consented in front of me as observer. Mentor/ 2nd mentee unobtrusive, but good introduction–in background but not "hidden"–properly introduced. Patient kept informed of* process.

*Mentee ensures patient is consented/comfortable with discussion. Discussion is interpreted for the patient.*

Great care was taken to ensure that the therapeutic relationship was preserved. This is of benefit for the patient as well as the mentee and their rights to that relationship were respected in the way that the cubicle was set up as well as the way in which the mentor unobtrusively observed the mentee and discussed their reasoning. In short, the patient played an essential role in the process, but the respect afforded them was not on this basis, but on the basis that normal patient care should be respected and the therapeutic relationship inviolable.

The patient being central to the process was also evident in the holistic questioning of the mentor. Before moving on to the mentee's learning and broader application, the patient's needs, concerns, diagnosis and management were thoroughly explored.

**OBSERVATIONAL NOTES (O4):** *Mentor provides more specific consideration / question, ensuring that patient's concerns are address*ed.

*Patient is central–thought given / empath*y

*Mentor focus*: *narrative reasoning, patient experience, hypothetico- deductive reasoning.*

The following notes were typical of the observations, showing how each area of the patient's experience, diagnosis and management were explored, ensuring patient-centred care. This was in line with the clinical reasoning model used to give structure to the mentoring [45].

**OBSERVATIONAL NOTES (O1):** *Mentor focuses on hypothetico-deductive and pattern recognition reasoning processes as well as bringing in some management reasoning early into the mentee's thoughts.*

*Mentor focus on narrative reasoning / patient's lived experience.*

*Mentor invites reflection with dual focus–the patient and the mentee.*

Pulling the mentor and mentee themes together and integrating the patient into the emerging model, illustrates how the different contributions of the mentor and mentee interact with one another, with a flow from the supportive environment, the mentor's provision and the mentee's contribution through to the patient (Fig 4).

*The clinical environment–providing the optimum context for development through MCP.* While the above analysis has explored the efforts made by the mentors to create a supportive environment for MCP, there are further environmental factors that emerged from the data,

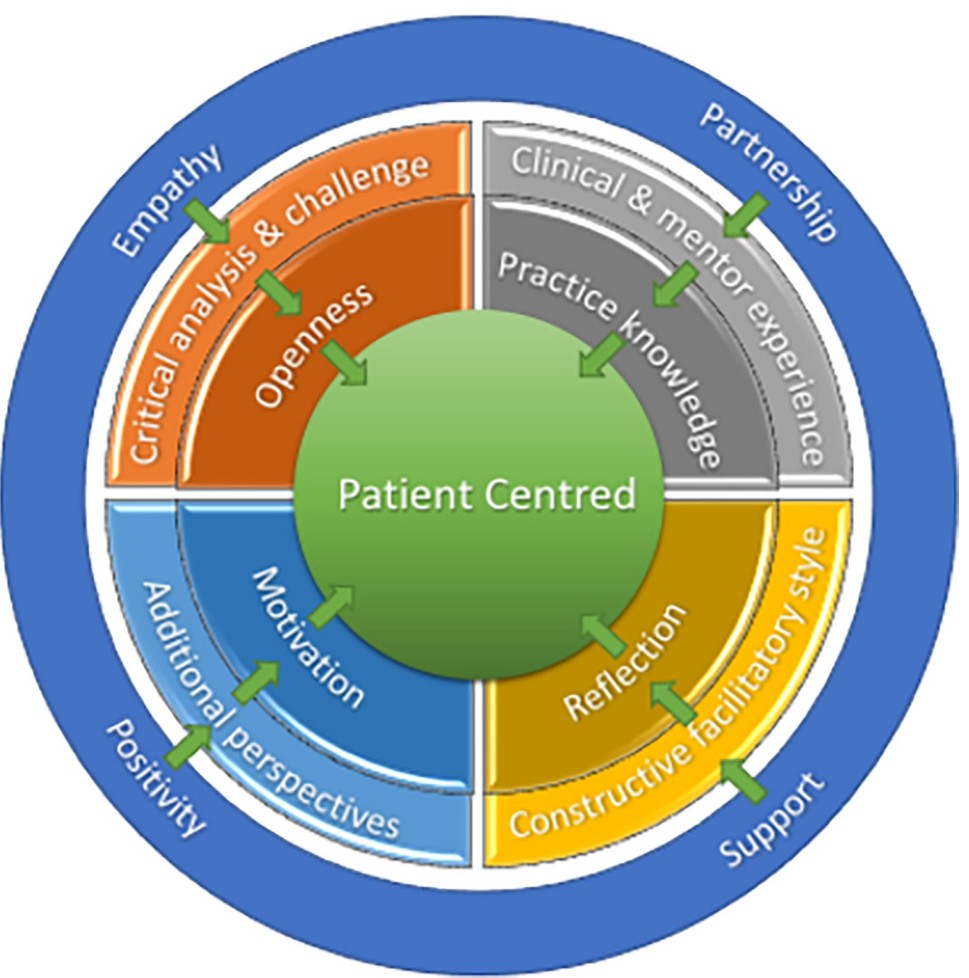

**Fig 4. Emerging model of MCP–integration of patient, mentee and mentor.**

with distinct advantages being identified relating to the fact that this process occurred in the mentee's usual clinical workplace.

*Environmental theme 1: Comfort/familiarity*: The fact that the intervention was intensive and sustained with high levels of challenge and requiring commitment to reflect from the mentees, made a familiar and comfortable environment attractive to the majority of mentees. The cognitive demand of the process was not distracted by being required to become familiar with new clinic contexts, colleagues and arrangements.

**Mentee 10:** *We were comfortable in the environment definitely because you know the department and don't have to worry about the runnings of the department or things like that. It was nice doing it with colleagues that you know. [340–346]*

*Environmental theme 2: Peer support*: The design of the trial rolled out the MCP intervention by department, meaning that mentees were not going through the process in isolation. This was seen as supportive in terms of the peer support that it offered:

**Mentee 1:** *Yeah, it was good to have your colleagues around during the process, going through the process with you. . . It would be a lot harder not to, I think. [657–658]*

This extended to times when the mentor was not present and augmented the mentee's reflection, which has already been identified as a critical factor for development:

**Mentee 11:** *Even though you may not have actually been doing the mentoring stuff that afternoon, if you'd spent the whole morning doing it you could then spend a bit of time reflecting outside of that process with other people... [270–273]*

*Environmental theme 3: Relationship with the mentor:* The fact that in the majority of cases the mentees had a prior relationship with the mentor, was identified as a helpful feature. Mentees identified that this prior relationship allowed them to be more open and honest in the discussions with mentors and facilitated their engagement with MCP more readily:

**Mentee 1:** *[It was] nice to do it with a relatively familiar face of a mentor and somebody you know...someone that you can talk to... you're familiar with them. [658–661]*

*Environmental theme 4: Time:* Central to MCP was allocating time for the mentor to observe the mentee's practice and also time to question, discuss and reflect together; the impact of this was emphasised by the mentees:

**Mentee 5:** *having that one to one time for somebody to give you constructive feedback... to process information and talk about why you didn't do something, why you did something, whereas usually 'Sign it off, I've got to get on to my next patient who's sat in the waiting room'. [51–55]*

The absence of time to reflect in a busy outpatient department where throughput is prioritised to meet efficiency targets is seen to be detrimental to reflection and in particular with patients with more challenging presentations:

**Mentee 5:** *I was... almost banging my head off a brick wall with this particular person, so we had time to sit down and (my mentor) said 'well you could have looked at it like this...' and we bounced ideas and that's where I found that... mentoring really useful whereas usually...I wouldn't have the chance to always reflect on what I've done. [85–91]*

Integrating all the themes together—from mentor, mentee, patient and environment—delivers a fully illustrated emergent model of the MCP process (Fig 5).

**Barriers to MCP.** In addition to the themes that emerged from the qualitative data on how the interactions between mentor, mentee and patient functioned with the aim of changing practice, themes also emerged on factors that potentially or actually negatively impacted the process. These themes related to the mentor, the mentee and the environment and should be seen as continuous rather than binary variables, in that they are factors that some mentees, mentors or environments may be more or less likely to present or may even vary in presenting them.

*The mentor: may bring external threats to reduce the impact of MCP.* **Mentor Barrier 1:** *Lack of relationship:* In the emergent model described above, the consensus was that already knowing the mentor facilitated the process rather than hindered it. One of the mentees found it difficult with the 2 of her 3 mentors that she didn't know prior to the process, particularly with reference to her overcoming her fear of being observed:

**Mentee 10:** *Yes, I think that goes back to the point about me feeling threatened by different experienced clinicians, mentors that haven't mentored me before, being judgemental on my*

*practice. That changed because we had different mentors a lot. So I had to go through that a few times before I got over it. . . [292–295]*

***Mentor Barrier 2**: **Excessively directive or overloading feedback***: A constructive facilitatory style emerged as a key factor in MCP, which did not preclude directives from the mentors, but these were given only after significant periods of mentor-guided mentee reflection. However, there were occasions where mentees felt that the directives were unhelpful in their development:

**Mentee 6:** *I'm happy to be pointed in different directions, whereas. . . now and then I think it was a little bit like trying to change my thoughts to suit their thought process which was a little bit of a barrier. [292–297]*

Interestingly the mentors were all self-aware on this point, being able to identify a tendency to revert to directives where mentee reflection was limited and attempting to correct this with stimulating reflection with further questions:

**Mentor 2:** *what I found really difficult about this [mentee] was getting self-reflection going and. . . I found myself resorting to telling them things, which I didn't really like, but I was aware that was happening. [36–40]*

**Mentor 3—***I think I do tend to overload people with feedback, with maybe too many smaller pieces of information that perhaps sometimes aren't that important. . . and then I would try and work in a bit of a team approach with the other mentors and just try to focus on pulling out the key things that would make the biggest difference to [the mentee]. [298–311]*

***Mentor Barrier 3**: **Differing emphases***: The additional perspective offered by the mentors was seen to be amplified by having multiple mentors as they gave different but complementary emphases in their facilitation of the mentees. However, these different perspectives were, on occasions, found by the mentees to be challenging:

**Mentee 1:** *The difficult thing was having mixed mentors with different approaches, although the underlying approach was similar. . . that variation amongst mentors was at times a little bit difficult to manage [585–589]*

While this was a definite barrier for some, others were able to rationalise this as natural differences between clinicians and felt able to process the differences in emphases as such:

**Mentee 11:** *No, I think I just reasoned it in my own head that everybody is different with maybe what they might bias or not bias and to what extreme I suppose. It didn't hinder, because ultimately knowing any other ways of doing things is a good thing because you can then decide what works a bit more with you as a clinician. [230–233]*

How these mentor barriers might impair different aspects of the emergent model is represented below (Fig 6).

*The Mentee*: *may bring internal threats to successful self-development*. **Barrier 1: *Defensive attitude***: In response to the critical analysis and challenge provided by the mentor, there were occasions when the mentees became defensive and disengaged with the process. Instead of a reasoned discussion about decision-making, the interaction became argumentative:

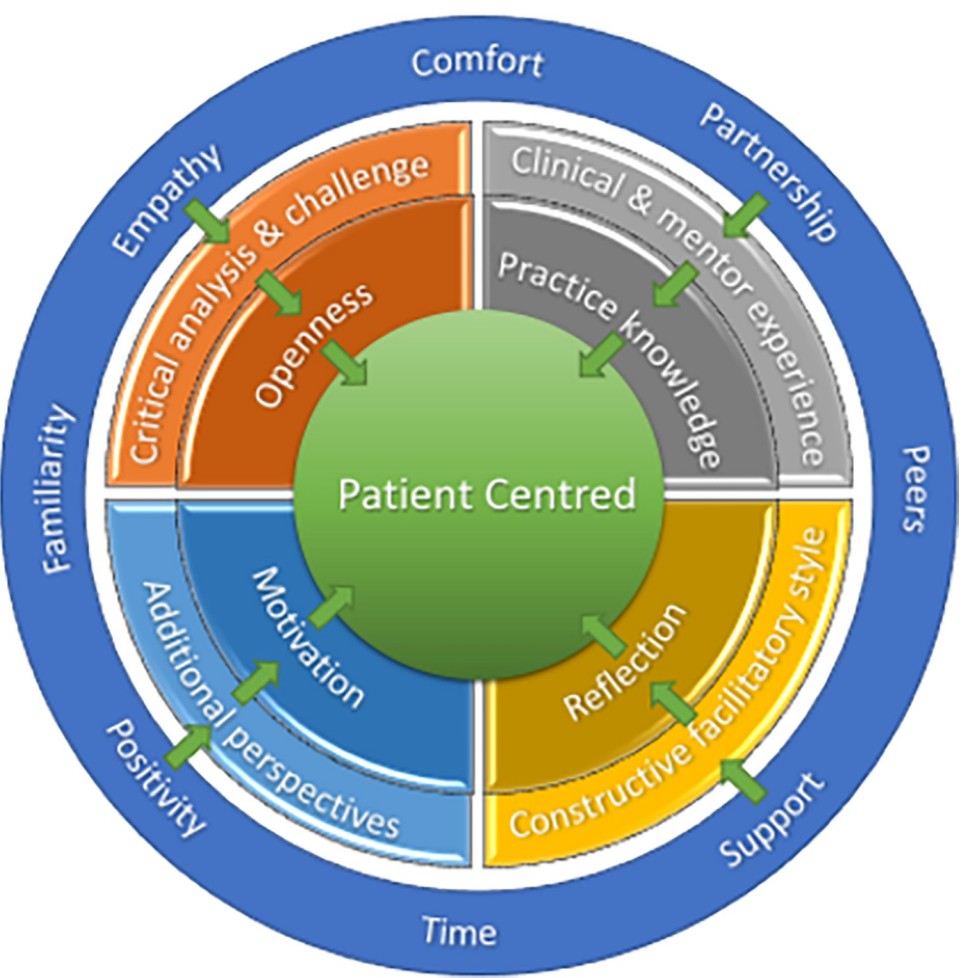

**Fig 5. The emergent model of MCP.**

**Mentee 5:** *I think it only happened once. . . but for want of a better word I threw my toys out of the pram and really reverted to type. 'This is what I'm seeing, this is what I think' and I probably wasn't willing to engage in conversation. I didn't want that to happen and maybe got a bit defensive and I think that defensiveness is potentially one personal barrier that I didn't want but it did happen. [286–290]*

While the reason for the conflict was not always identified, mentees were able to reflect on factors such as tiredness and insecurity being the triggers for such defensiveness. They were also able to show how these barriers were quickly dismantled by confronting the issue rather than avoiding it, thus restoring the mentor-mentee relationship:

**Mentee 10:** *I actually sat down with the mentor at the time and said this. . . was detrimental at the time because I was closing myself to not accepting what things were being said and using it as a development tool. . . and not being able to open up and learn these things. [258–264]*

***Barrier 2: Routine working / admitting change was needed:*** The mentor's expectation for the mentee to reflect on the feedback they have been given and to apply it to practice was

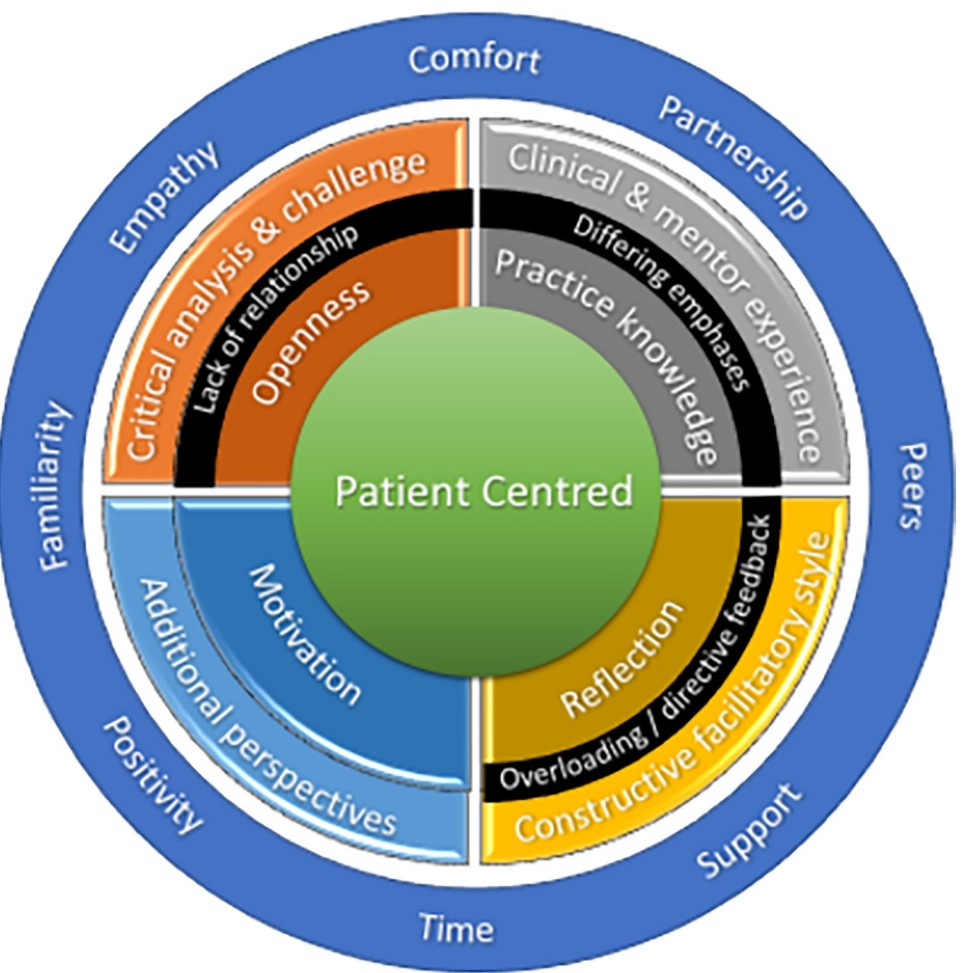

**Fig 6. Mentor barriers to the emergent model of MCP.**

central to MCP transforming the mentee's practice. A mentee entering the process desiring practice validation rather than being motivated to transform practice is likely to receive circular or repetitive feedback as the same issues are highlighted as being evident. This can lead to frustration for the mentor:

> **Mentor 3:** *It's quite important that they take on the information that you have reflected upon together and they integrate it and practice it daily, so they can build on it for the next time, rather than saying the same thing time and time again because. . . they don't want to make that change. [54–57]*

The issue of routine working is one that the mentees could identify as being problematic and mentors were very familiar with trying to overcome and were able to identify typical underlying reasons why clinicians can fall into this pattern of working:

> **Mentee 2 (talking about barriers to progress)**–*. . .doing things because you've always done them. . .that type of way of working where you know something works and so you do use that particular technique for lots of people. [401–404]*

*Mentor 2 (discussing a mentee)*—*He's had patient mileage which is good and helpful. . . I think that did have a bit of a flip side in that there was a tendency to go 'I've seen this and done it before, this is what you do here' rather than 'I'm going to think this individual through'.* [248–251]

***Barrier 3***: ***People pleasing or performing***: An unexpected theme emerged from the data as several mentees honestly reflected that, on occasions, they found themselves 'performing' in ways that they thought would please their mentor. Ultimately, this is a superficial and likely transient change to practice rather than the deeper more substantive transformation that results from genuine reflection and application.

**Mentee 10:** *I almost felt like sometimes we were doing things or saying things to please them or the way they do things or think about things, rather than developing ourselves.* [306–307]

The problem of seeking to behave in a way that pleases the mentor was seen to depart from the ideal of using the mentee's current practice as foundational and building on strengths and developing areas of weakness. Instead, by performing for the mentor's approval, the mentee experienced a lack of form or structure for practice change:

**Mentee 1:** *I think if you're keen to change and want to please a mentor and show that you can do something differently. . .if that's a side to your personality that you have during a learning experience which I seem to have. . .you end up losing your whole basis and grounding of your assessment.* [525–532]

***Barrier 4***: ***Lack of knowledge / exposure / previous negative experience***: The mentee's clinical knowledge and experience and previous exposure to observed practice was seen as foundational for MCP; the absence of such aspects emerged as a barrier to it. Mentors, identified the lack of knowledge of key concepts and a lack of previous exposure to observed practice as challenging to MCP:

**Mentor 2:** *'Why are we getting you to do all this clinical reasoning all of a sudden? Why are we talking about collaboration? Why are we talking about pain mechanisms?' There seemed to be quite a lot of, maybe not alien concepts, but concepts we were introducing, quite big concepts, there was not a foundation of theoretical knowledge about those things.* [193–197]

**Mentor 2:** *[When] watched assessments are the norm, it's expected so therefore. . .it's part of the culture so you accept it and get on with it. All of a sudden we are doing something dramatically different so I felt it was almost a bit of a shock and [the mentee] didn't know how it was going to work, he didn't know what we were going to do.* [185–191]

However, it was not just lack of exposure to observed clinical practice that was a barrier. One mentee, reflecting on a mentoring relationship in a previous place of employment, found that her previous exposure to observed practice was a highly negative experience that she had to overcome in engaging with this process:

**Mentee 3:** *I worked with a colleague that wasn't. . . the nicest mentor, so I think that freaked me a little bit and challenged me. . .negatively a little bit. So, I think it took me a while. . .before I felt I wanted to do it again, if that makes sense. It didn't scar me. . .but then I presume it did do because I didn't want to do it.* [455–460]

*Barrier 5*: *Fear of being exposed/looking weak*: A potentially surprising theme that emerged from the data was that of fear being the main barrier to engaging with the process at the outset; even though all the mentees were volunteers, it was a common finding that mentees felt that they had to overcome this barrier above all. What comes out of the data helps to understand the mentee's perspective in being observed and critiqued was seen as potentially threatening to their professional identity:

**Mentee 11:** *Being terrified that you might be exposed that you're not good enough. . . I didn't want somebody to turn around and say 'well she's not that good' which is ridiculous but yes, that was definitely a huge thing for me. [183–186]*

In the physiotherapy clinic, individual clinicians will have their own caseload and treat patients in individual cubicles. Inviting another clinician into this context and opening oneself up to critique allows the mentor to see the actuality of practice with its strengths and weaknesses. Not only was this felt as exposing, but the very act of inviting in a mentor was seen to be admitting weakness:

**Mentee 10:** *I was more afraid of what people were thinking about me and my practice, rather than using it to learn. I think that was a big thing for me to get over in the beginning. . . it was that fear of being judged for how you practice. [264–267]*

The mentors were cognizant of this fear and were empathetic to the mentees, explicitly and skilfully seeking to address and overcome this fear through different strategies:

**Mentor 2:** *a bit of a barrier was the threat of someone external coming in and professional pride and not wanting to make yourself vulnerable there and that was overcome by him watching us. [202–205]*

These mentee barriers as they might impair different aspects of the emergent model are represented below (Fig 7).

*The Clinical Environment*: *has potential to undermine development through MCP*. The final themes that emerged as barriers to the process all relate to the clinical environment. While the mentees found the comfort and familiarity and peer support that was offered to be helpful, there were hindrances identified from undertaking MCP in their own workplaces.

*Environmental barrier 1*: *Time encroachment/distraction*: The mentees diaries were set up to have protected sessions with the mentor that allocated appropriate discussion time after the patient interaction. However, outside of these sessions the mentees had a clinical or managerial caseload to attend to which proved challenging to balance and was identified as having a negative effect on the mentee being able to be single-minded in their focus on the process:

**Mentee 3:** *. . .it was very much kind of, you know "I'm doing my mentoring sessions" but then there'd be a problem. I think the main thing was because I was here in my own environment, that I couldn't escape the everyday stuff that was going on. [574–596]*

These pressures led to mentees occasionally booking additional patients into MCP time which was identified as counter-productive to reflection and self-critique:

**Mentee 4:** *I think initially I found it very difficult partly down to the fact that we didn't necessarily have the. . . time for reflection in the first one or two weeks. [543–546]*

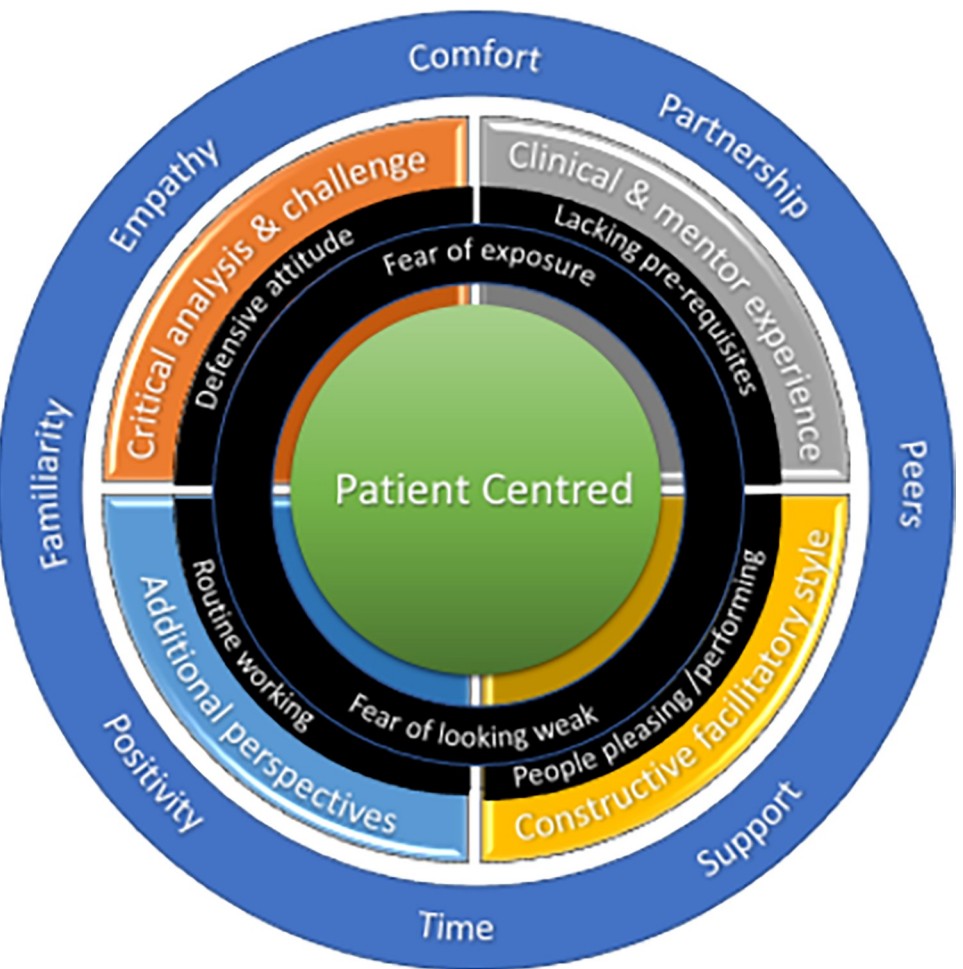

**Fig 7. Mentor barriers to the emergent model of MCP.**

*Environmental barrier 2*: *Peer presence*: While peer presence was largely identified as a supportive influence, there were exceptions to that view. The practice of delivering MCP on a 2 mentee to 1 mentor basis meant that mentees often had their peers present and involved in the discussions with the mentor which some mentees found challenging:

**Mentee 1:** . . .*us sitting in with each other and then feeling 'How do we go about joining in with this constructive criticism without sounding like we're sitting on a high horse?' saying 'Yeah, I wouldn't do that myself', it's a very difficult line. . . [627–630]*

In addition some mentees found that their peers in the department who were not engaged in the process (despite being afforded the same opportunity to volunteer) put additional pressure on them, being critical of the time and input that was given to the mentee that they were not enjoying:

**Mentee 9:** *Yes, they were being critical of the process and critical that they weren't having it and I was and other people being critical of it because they were having to do more than they normally would. So, the environment for me was massive. . . [1038–1040]*

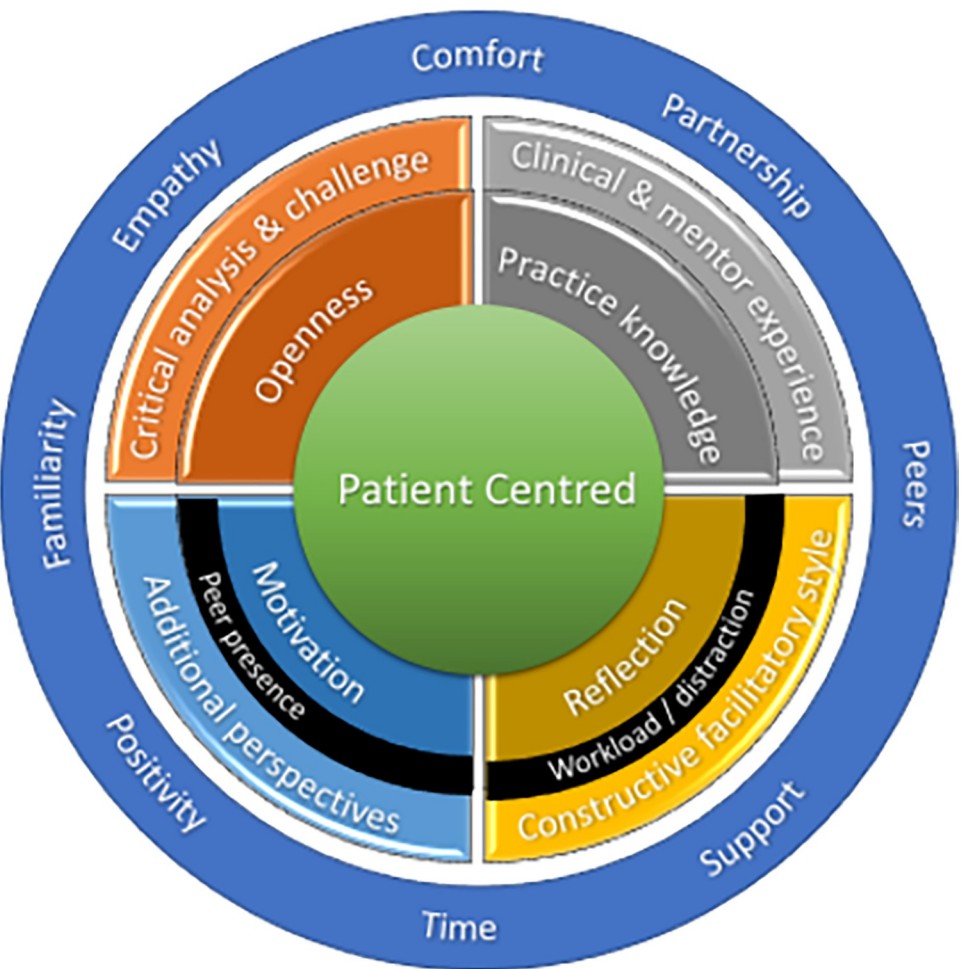

**Fig 8. Environmental barriers to the emergent model of MCP.**

These environmental barriers are represented as they may impair different aspects of the emergent model below (Fig 8).

## Discussion

### Findings

The objectives of this study were to explore mentor and mentee perceptions of a mentored clinical practice programme—that demonstrated significant improvement in patient outcomes—in order to identify key factors in the process to apply to future clinicians and so benefit their patients. Findings from the data from semi-structured interviews and observations of the mentoring process identified key enablers and barriers to the complex interaction between mentor, mentee, patient and environment as illustrated in the emergent model of MCP.

The key enablers were mentors drawing on their clinical and mentoring experience to provide additional perspectives and critical analysis and challenge to mentees in a constructive and facilitatory style. This was augmented by mentees being motivated and open to receive critique to reflect on and develop their practice knowledge. This mentoring process worked best when the environment was familiar, comfortable, with peer support and with adequate time

allocated for reflection and discussion and when the relationship fostered was an empathetic, supportive, positive partnership.

The key barriers identified were mentors being overloading or too directive with feedback, differing emphases across mentors and a lack of relationship between mentor and mentee. Key mentee barriers were a defensive attitude, a lack of pre-requisite knowledge, a desire to continue routine working or to perform to please the mentor and fear of looking weak or being exposed as being substandard. Environmentally, peer presence or criticism could be a barrier as well as time for reflection and discussion being removed.

**The emergent model of MCP.** MCP has been proposed as being effective through providing the clinician with feedback that is accurate, immediate and specific [46–48]. The findings of this study support this proposal in highlighting the importance of the additional perspective delivered by the mentor. This study substantiates previous claims and findings of the value of the mentor's ability to stand back from the situation and see the interaction as a whole [48] and guide the clinician on their journey towards expertise [49] with the clinicians in receipt of this perspective—and mentors providing it—identifying it first-hand as a powerful component of learning and development. Furthermore, the mentor's provision of a dual focus on patient and mentee, identifying mentee learning needs through sustained, careful observation of the motivated mentee's practice and the amplification of perspectives through the use of multiple mentors are all new insights that this study brings.

The mentors' critical analysis of the participants' practice knowledge is congruent with a previous study in MSK physiotherapy where a mentored clinical practice module was identified as the most powerful learning experience for practice change on a Masters programme, with critical dialogue and evaluation providing a "contradiction" experience [24, 25]. The theme of the mentor's provision of critical analysis and challenge mirrored by the theme of mentee openness in this study confirm these assertions, with the data giving further insight into the mentor's perspectives and activities. The mentor provided challenge by facilitating the mentee to provide evidence to justify their decision-making, testing and questioning (reflecting findings on the importance of justification in masters level practice [21]).

This critique was aimed at addressing identified gaps in the mentee's reasoning and was delivered in a constructive manner with intentionally varied levels of critique as the mentors were mindful of the mentee's capacity to accept criticism, allowing the mentees to observe and critique them in turn. The critique was balanced and prioritised to areas that would elicit practice change most readily.

The finding of the mentors' constructive and facilitatory style is substantially a new contribution to the literature, with detail from observations of the MCP process and mentor interviews added to the perspectives of the participants. The mentor's critique was aimed at addressing identified gaps in the mentee's reasoning and was delivered in a constructive manner with intentionally varied levels of critique as the mentors were mindful of the mentee's capacity to accept criticism, allowing the mentees to observe and critique them in turn. The critique was balanced and prioritised to areas that would elicit practice change most readily. The support offered by mentors has been seen as essential to offset the high levels of challenge. It has been suggested previously that mentors may require guidance to offset the high levels of challenge [25] in an environment that is collaborative and respectful to support learning, given the power relationship between the mentor and the mentee [48]. The mentors in this study were deliberate in their actions to avoid a hierarchical approach; this humility displayed by mentors (being open to be observed and critiqued themselves and removing hierarchical barriers) and mentees (being aware of their limitations and inviting practice challenge) is reminiscent of studied experts in physiotherapy practice, where humility was a key trait [50].

The key theme of mentee reflection stimulated by the mentor's facilitatory style evidenced in this study has a strong theoretical construct in the literature. The ability of masters students to reflect in and on action and self-evaluate in order to develop identified traits of expertise [35] such as recognising their limitations and developing metacognition have been identified to be key characteristics for masters level practice [21, 51]. The assertion that when practitioners embrace a contradictory clinical experience they are more likely to engage in critical reflection and their practice is more likely to adapt has been proposed as well as its counterpart: contradictory clinical experiences may be rejected by the practitioner with little change to practice likely to result [46, 52, 53]. As such, it has been proposed that a practitioners' ability to learn from experience hinges on the degree to which they are open to change their practice, exposed to alternative perspectives and ready to engage in reflection [54]. The current study provides direct evidence of this, as mentors were observed providing alternative perspectives and questioning to provide challenge and contradiction to stimulate the mentees' reflections. Questioning always began with mentee reflection with mentors providing structure to the discussion and organisation to the mentee's thoughts before any directive feedback was given. The facilitation also went from the specific (the immediate patient) to the general–the mentee's reasoning skills and future patients) and incorporated a pattern to allow mentees to continue to self-reflect beyond the intervention.

This study provides new information on the supports of undertaking such an educational process in the usual workplace of the clinician. It has been previously claimed that supportive clinical learning environments can contribute to improved patient care through direct effects on clinicians [55]. Learning in the workplace environment has been argued to be an interdependent process of opportunity and engagement that shapes learning, the workplace providing support as well as imposing constraints on individuals as they make decisions on how they will participate in—and what they learn from—what they experience [56, 57]. This study supports this argument as the environmental supports afforded to the participants (as well as the constraints placed upon them) were clearly identified by mentors and mentees as well as identifying the subsequent impact on their engagement with the process. This study adds findings on key aspects of a supportive environment, with the mentor providing personal support, empathy and positivity in a relationship of partnership within a comfortable, familiar context supported by peer presence and time for reflection and analysis.

**Overcoming barriers.** *Mentor barriers*. The findings of this study provide clinicians with an awareness of potential problems and possible solutions in mentoring relationships. Existing literature has previously found that where support was not forthcoming from a mentor, strong negative emotions were engendered in the mentee [25]. The findings of this study go further by exploring the mentors' perspectives together with those of the mentees. Working with mentors that mentees already had prior working relationships with was mostly seen as supportive but could unhelpfully impact on the mentoring dynamic as pre-existing relationships changed in this new context of intensive MCP. Mentors and mentees identified the pitfall of mentors becoming overly or prematurely directive with feedback, thus bypassing the deep reflective processes they were attempting to stimulate in the mentee. While multiple mentors developing the mentee was strongly supported in that the additional perspectives provided gave a broader, more rounded experience to the mentees, it introduced the danger of feedback being confusing or even contradictory. Mentors attempted to overcome this by communicating with each other and developing a consensus of what to cover as a priority for their mentees.

*Mentee barriers*. It has been argued that MCP limits against defensive responses as the critical appraisal by the mentor enables the mentee to become aware of their practice knowledge which might otherwise be hidden [24, 25]. However, in this study, revealing hidden practice knowledge did not automatically prevent mentees being defensive against receiving, accepting or engaging with critical analysis and feedback given by the mentors. Mentees identified

tiredness and insecurity leading to argumentation, anxiety and closing down to suggestions, thus reducing the learning opportunities afforded.

Routine working where practitioners perform examination and management approaches routinely, regardless of the patient presentation [46, 48, 58] has been postulated to result in practitioners' experience being circular in nature [59] limiting the ability to learn in and from practice [48]. A move away from routine working was described by physiotherapists on a masters level programme where mentored clinical practice was employed [21]. This study reveals that mentors saw routine working as revealing a lack of mentee motivation for receiving the additional perspectives that they were able to provide. Mentees were also able to identify how their routine working practices were challenged by the mentors in the process, particularly as they were asked to justify their reasoning and decision-making.

The mentee barrier of fear has been proposed previously, with workplace observation being perceived as a monitoring mechanism [60] resulting in feelings of anxiety and vulnerability [48] and fears of harsh, negative judgment of their clinical practice, or loss of respect [48]. Mentees faced with the prospect of mentors closely scrutinizing their practice on a Masters programme expressed feelings of being 'terrified' and 'scared to death' [25]. These emotions were reflected in this study as mentees fear was evidenced heavily, centred around the threat to the professional identity of the mentee in looking weak at admitting the need to change and in being exposed as sub-standard in their practice. The suggestion that mentees' fears can be allayed by the mentor providing constructive feedback and support [25] was evidenced in this study along with further perspectives on fear. Mentors were cognizant of this barrier and empathetic to the mentee and overcame it by being supportive and making themselves vulnerable by inviting the mentees to observe and critique their practice. This mirroring of behaviour and humility of the mentors and mentees reflects proposals in the literature that the success of an observed practice intervention depends on the process being collaborative, respectful and supportive and driven by genuine desires to facilitate learning and expertise [48].

The theme of people-pleasing and performing is new to this literature, but several mentees found themselves falling into the trap of performing to fit with the mentor's preferences. Superficial performance to gain a mentor's approval risks genuinely achieving profound and lasting practice change and interestingly, it was the mentees who were able to identify this as a barrier with self-awareness previously being identified as essential to reflective practice [46]. Furthermore, the mentees felt secure enough in their relationship with the mentor to admit it and discuss it explicitly. This supports the importance of the mentor and mentee making every effort in observed practice contexts to develop self-awareness and critique as well as a trusting, supportive and constructive relationship.

The evidence of the barrier of the mentee's practice knowledge being insufficient for the process, or the mentee having no–or negative–previous experiences of observed practice raises questions. Is MCP only for those with experience, being less likely to be effective for newly qualified practitioners? In the UK in occupational therapy [61–63] informal feedback from colleagues with greater experience that was challenging but enhanced by explicit reassurance, was reported as being highly beneficial for guiding practice and building practitioners' confidence and commitment to learning. The findings of this study relate to the intensity of MCP and the level of the questioning and challenge. Delivering MCP to newly qualified or undergraduate physiotherapists is likely to require a lower level of questioning and intensity initially, with the mentor adjusting their expectations and goals in collaboration with the mentee.

*Environmental barriers.* Workplace environments suggested to stifle learning are those that place an emphasis on efficiency and throughput, limiting time for practitioners to critically reflect on clinical experience [46, 64], or those that promote lone working contexts and limit opportunities for practitioners to work together and learn from each other [65]. As a result,

clinicians are unlikely to reflect deeply on their reasoning behind their actions and so develop automatic, habitual procedures, techniques and strategies that are routinely applied to patients [46, 54, 66]. This was confirmed by the participants of this study where even though time was allocated for discussion, reflection and feedback, the pressures of the workplace impacted upon that time either directly, by being called away to deal with situations, or indirectly, by being cognitively distracted or emotionally anxious. These were tangible disadvantages of implementing this type of intervention in the participants' usual work context. Similarly, peer presence was not universally a positive support, where jealousy or criticism became demotivating to participants who struggled with feelings of guilt at not contributing as much to the day to day running of the department.

## Study limitations and further work

It is important to acknowledge that this study occurred in one NHS organisation at a particular time and under particular circumstances and so only limited generalisation is warranted on that basis. On reflection the theme of the patient perspective is the least developed. While qualitative observations revealed the patient's place and role in MCP, the absence of any interview data with patients felt obvious during reflections on the analysis. Both mentor and mentee observational data was able to be integrated with interview data which was unavailable for the patient perspective; giving the patient voice through semi-structured interviews would have added insight, strengthening the patient data and deepened the researchers' understanding of the process. Future work could explore the patient voice in addressing this, as well as exploring incorporating patient feedback to the mentee as part of a mentored clinical practice.

## Conclusions

This study has been able to explore processes in the MCP programme that delivered highly significant improvement in patients of MSK physiotherapists and to identify key themes which are central to the intervention. These key themes that emerged were evidenced from the rich data provided by qualitative interviews with the clinically mentored staff and their clinical mentors and observations of MCP. A model emerged from these themes to demonstrate how they interacted in order for MCP to be optimally effective. Performing MCP in the mentees' usual clinical environment provided familiarity, comfort and peer support for the mentees, along with the time necessary to discuss and reflect on their reasoning. The mentees found the relationship with their mentors to be supportive, positive and empathetic employing a partnership approach which augmented this supportive environment. The mentees were motivated to receive the additional perspectives brought by their mentors and were open to the critical analysis and challenge to their practice that the mentors brought. The mentors' constructive facilitatory style stimulated the mentees to reflect critically on their own practice as the mentors' clinical knowledge and previous experience of delivering MCP allowed the mentees to challenge their own practice knowledge building on the foundations of their own experience. All of this was channelled to the patient who was central to the process, benefitting from the combined reflections and discussions of mentor and mentee. The emergent model also evidences risks to the process which mentors and mentees should confront and address to ensure MCP is optimal for patient care–and patient outcomes—to be impacted.

## Acknowledgments

We thank the patients who consented to be part of the mentoring process and the physiotherapists and their mentors who committed to this study. We thank the Cardiff and Vale University Health Board for their support in allowing the study to run in their physiotherapy service.

We are grateful to the Musculoskeletal Association of Chartered Physiotherapists and to Cardiff University for covering the open access fee for publication.

## Author Contributions

**Conceptualization:** Aled Williams, Ceri J. Phillips, Alison Rushton.

**Data curation:** Aled Williams.

**Formal analysis:** Aled Williams.

**Investigation:** Aled Williams.

**Methodology:** Aled Williams, Alison Rushton.

**Project administration:** Aled Williams, Ceri J. Phillips.

**Supervision:** Ceri J. Phillips, Alison Rushton.

**Writing – original draft:** Aled Williams.

**Writing – review & editing:** Ceri J. Phillips, Alison Rushton.

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
