## [Decision Letter · Decision Letter 0]

1 May 2022

PONE-D-21-35299Exploration of mentor and mentee perspectives of a mentored clinical practice programme to improve patient outcomes in musculoskeletal physiotherapyPLOS ONE

Dear Dr. Williams, 

Thank you for submitting your manuscript to PLOS ONE. After careful consideration, we feel that it has merit but there are areas where it can be improved further. Therefore, we invite you to submit a revised version of the manuscript that addresses the points raised during the review process.

We look forward to receiving your revised manuscript.

Kind regards,

Soham Bandyopadhyay

Academic Editor

PLOS ONE

Journal Requirements:

Reviewers' comments:

Reviewer's Responses to Questions

**Comments to the Author**

1. Is the manuscript technically sound, and do the data support the conclusions?

Reviewer #1: Yes

Reviewer #2: Yes

2. Has the statistical analysis been performed appropriately and rigorously? 

Reviewer #1: Yes

Reviewer #2: N/A

3. Have the authors made all data underlying the findings in their manuscript fully available?

Reviewer #1: Yes

Reviewer #2: No

4. Is the manuscript presented in an intelligible fashion and written in standard English?

Reviewer #1: Yes

Reviewer #2: No

5. Review Comments to the Author

Reviewer #1: Titile is well explained and author has used easy english to make it understanable.

Topic is new and helpful in filed of physiotherapy.

Abstact is well explained with limited words.

Introduction is well written in proper english language and Author has explained the objective of study thoroughly.

All factors of Methodology were explaiend in proper but few points need more concentration.

Data analysis and result is explaied.

Discussion is well explained incliding all the points to be focused on.

conclusion is intrepeted properly

Reviewer #2: This study aims to explore mentor and mentee perceptions of a mentored clinical practice programme and identify key factors in the process to improve patient outcomes. While the paper in many ways is well written, it has some weaknesses that need to be addressed before eventual publication.

Line 51-53: I am not too sure about the advantage of using so many citations for a single argument/statement. Authors are advised to keep relevant citation/s only.

Line 55-60: The first sentence describing the results is not necessary when you have interpreted the results in the second sentence. The paragraph should also have a concluding remark.

Line 65: Please explain what is M level.

Line 83-98: It is difficult to understand which section describes the methods employed in this particular study. Please revise.

Line 110: Please explain what are these band levels.

Line 213-214: Please check for grammar.

Line 819: Please remove.

Line 821-831: The discussion section should start with reiterating the objectives and key findings of the study.

Quotations: All quotations should have line numbers given in the transcripts.

The manuscript is quite long. The results section should provide succinct findings and explanations/interpretations. Also, 1 quotation/observation note per theme should be enough.

6. PLOS authors have the option to publish the peer review history of their article (what does this mean?). If published, this will include your full peer review and any attached files.

Reviewer #1: **Yes: **ifra aman

Reviewer #2: No

---

## [Author Response · Author response to Decision Letter 0]

14 Jun 2022

Line 51-53: I am not too sure about the advantage of using so many citations for a single argument/statement. Authors are advised to keep relevant citation/s only.

Response: The point being made is the debate of this point over several decades, hence multiple sources being cited to support that multiple authors have been making this call over the decades.

Line 55-60: The first sentence describing the results is not necessary when you have interpreted the results in the second sentence. The paragraph should also have a concluding remark.

Response: Sentences edited and merged with concluding remark inserted for the paragraph.

Line 65: Please explain what is M level.

Response: explained in text.

Line 83-98: It is difficult to understand which section describes the methods employed in this particular study. Please revise.

Sentences revised and edited to clarify the methods employed.

Line 110: Please explain what are these band levels.

Band levels explained in text with additional reference.

Line 213-214: Please check for grammar.

Checked and corrected

Line 819: Please remove.

Removed

Line 821-831: The discussion section should start with reiterating the objectives and key findings of the study.

Initial paragraph inserted into the manuscript for this purpose.

Quotations: All quotations should have line numbers given in the transcripts.

The manuscript is quite long. The results section should provide succinct findings and explanations/interpretations. Also, 1 quotation/observation note per theme should be enough.

Quotation line numbers inserted. Multiple quotes removed to leave (usually) 1 quote per theme/subtheme to significantly shorten the length of the article.

---

## [Decision Letter · Decision Letter 1]

19 Jul 2022

PONE-D-21-35299R1Exploration of mentor and mentee perspectives of a mentored clinical practice programme to improve patient outcomes in musculoskeletal physiotherapyPLOS ONE

Dear Dr. Williams,

Thank you for submitting your manuscript to PLOS ONE. After careful consideration, we feel that it has merit but does not fully meet PLOS ONE’s publication criteria as it currently stands. Therefore, we invite you to submit a revised version of the manuscript that addresses the points raised during the review process.

We look forward to receiving your revised manuscript.

Kind regards,

Soham Bandyopadhyay

Academic Editor

PLOS ONE

Journal Requirements:

Reviewers' comments:

Reviewer's Responses to Questions

**Comments to the Author**

1. If the authors have adequately addressed your comments raised in a previous round of review and you feel that this manuscript is now acceptable for publication, you may indicate that here to bypass the “Comments to the Author” section, enter your conflict of interest statement in the “Confidential to Editor” section, and submit your "Accept" recommendation.

Reviewer #1: All comments have been addressed

Reviewer #2: (No Response)

2. Is the manuscript technically sound, and do the data support the conclusions?

Reviewer #1: Yes

Reviewer #2: Yes

3. Has the statistical analysis been performed appropriately and rigorously? 

Reviewer #1: Yes

Reviewer #2: N/A

4. Have the authors made all data underlying the findings in their manuscript fully available?

Reviewer #1: Yes

Reviewer #2: No

5. Is the manuscript presented in an intelligible fashion and written in standard English?

Reviewer #1: Yes

Reviewer #2: No

6. Review Comments to the Author

Reviewer #1: All the questions were answered properly.

Data was analysed properly and written in a sounded fashionable way which matches it conclusion.

Reviewer #2: Line 84-95: Please describe your own study design NOT what has been employed in previous studies. You can add a separate paragraph on previous studies to support your methodology.

This comment also applies to participation selection and data collection and setting sections.

Line 755: Why the heading 'discussion' has been deleted?

Line 759-762: Please add key findings of the study.

7. PLOS authors have the option to publish the peer review history of their article (what does this mean?). If published, this will include your full peer review and any attached files.

Reviewer #1: **Yes: **ifra aman

Reviewer #2: No

---

## [Author Response · Author response to Decision Letter 1]

21 Jul 2022

Line 84-95: Please describe your own study design NOT what has been employed in previous studies. You can add a separate paragraph on previous studies to support your methodology.

This comment also applies to participation selection and data collection and setting sections.

Response: The text has been edited to state the study design, participant selection, data collection and settings at the beginning of the paragraphs before giving the justification for these decisions.

Line 755: Why the heading 'discussion' has been deleted?

Response: Heading has been reinserted.

Line 65: Please explain what is M level.

Response: explained in text.

Line 759-762: Please add key findings of the study.

Key findings have been added and summarised prior to discussion paragraphs.

---

## [Editor Report · Decision Letter 2]

26 Jul 2022

Exploration of mentor and mentee perspectives of a mentored clinical practice programme to improve patient outcomes in musculoskeletal physiotherapy

PONE-D-21-35299R2

Dear Dr. Williams

We’re pleased to inform you that your manuscript has been judged scientifically suitable for publication and will be formally accepted for publication once it meets all outstanding technical requirements.

Kind regards,

Soham Bandyopadhyay

Academic Editor

PLOS ONE

---

## [Editor Report · Acceptance letter]

10 Aug 2022

PONE-D-21-35299R2 

Exploration of mentor and mentee perspectives of a mentored clinical practice programme to improve patient outcomes in musculoskeletal physiotherapy 

Dear Dr. Williams:

I'm pleased to inform you that your manuscript has been deemed suitable for publication in PLOS ONE. Congratulations! Your manuscript is now with our production department. 

Kind regards, 

on behalf of

Dr. Soham Bandyopadhyay 

Academic Editor

PLOS ONE